# Factors Associated with White Fat Browning: New Regulators of Lipid Metabolism

**DOI:** 10.3390/ijms23147641

**Published:** 2022-07-11

**Authors:** Peiwen Zhang, Yuxu He, Shuang Wu, Xinrong Li, Xutao Lin, Mailin Gan, Lei Chen, Ye Zhao, Lili Niu, Shunhua Zhang, Xuewei Li, Li Zhu, Linyuan Shen

**Affiliations:** 1College of Animal Science and Technology, Sichuan Agricultural University, Chengdu 611130, China; zpw1995@stu.sicau.edu.cn (P.Z.); h2021202015@stu.sicau.edu.cn (Y.H.); wushuang@stu.sicau.edu.cn (S.W.); 2020302147@stu.sicau.edu.cn (X.L.); 202000576@stu.sicau.edu.cn (X.L.); ganmailin@stu.sicau.edu.cn (M.G.); chenlei815918@sicau.edu.cn (L.C.); zhye@sicau.edu.cn (Y.Z.); niulili@sicau.edu.cn (L.N.); 14081@sicau.edu.cn (S.Z.); xuewei.li@sicau.edu.cn (X.L.); 2Farm Animal Genetic Resources Exploration and Innovation Key Laboratory of Sichuan Province, Sichuan Agricultural University, Chengdu 611130, China

**Keywords:** white fat, brown fat, obesity, epigenetic, fat thermogenesis

## Abstract

Mammalian adipose tissue can be divided into white and brown adipose tissue based on its colour, location, and cellular structure. Certain conditions, such as sympathetic nerve excitement, can induce the white adipose adipocytes into a new type of adipocytes, known as beige adipocytes. The process, leading to the conversion of white adipocytes into beige adipocytes, is called white fat browning. The dynamic balance between white and beige adipocytes is closely related to the body’s metabolic homeostasis. Studying the signal transduction pathways of the white fat browning might provide novel ideas for the treatment of obesity and alleviation of obesity-related glucose and lipid metabolism disorders. This article aimed to provide an overview of recent advances in understanding white fat browning and the role of BAT in lipid metabolism.

## 1. Introduction

Improved living standards have increased the number of obese and overweight people. Obesity is a risk factor for hyperlipidaemia, hypertension, and other diseases [1,2]. The potential risk factors for obesity not only include body fat contents, but also the type and distribution of fats. For example, some people do not show the signs of metabolic dysfunction, despite having an obese phenotype. According to the metabolic point of view, such people are considered “healthy” obese [3]. Therefore, it is not the amount of fat itself, which causes its associated diseases, but rather its positioning and individualized qualitative functional characteristics.

Adipose tissue plays an important role in energy metabolism [4] and endocrine organs [5] in the human body. The existing studies classify fats into three types based on the differences in their function and structure [6], including white adipose tissue (WAT), brown adipose tissue (BAT), and controversial beige fat cells [7]. Brown adipose tissue, as compared to the energy-storing WAT, can generate heat, thereby consuming calories for the body [8]. Many studies have shown that under certain conditions, stimulating WAT could promote its conversion to BAT [9]. As obesity is a metabolic disease caused by the excessive lipid deposition in WAT, researchers are focusing on boosting its energy metabolism. One method of treating metabolic diseases, such as obesity, is to promote the browning of WAT. This might provide a potential therapeutic method for the treatment of obesity [10], polycystic ovary [11], and other metabolic syndrome and related disorders. At present, numerous studies have shown that different factors play a role in regulating the WAT browning. This review has summarized the effects of regulatory factors and WATB methods.

## 2. Types and Functions of Fat

Obesity is negatively associated with the increase in social productivity and endangers people’s health [12]. It is caused by the excessive energy consumption [13] as well as the collection and an excessive level of triglycerides and lipids, which are stored in adipocytes [14]. Furthermore, adipose tissues are the body’s most important metabolic and endocrine sites as well as the primary energy storage sites. Therefore, researchers are deeply concerned about fat formation and metabolism [15].

Fat is a tissue and an organ with secretory functions and has been roughly divided into white fat, brown fat, and ectopic deposited fat [16]. Hormones can be secreted by adipose tissue [17], regulatory factors [18], and exosomes [19], acting on other related organs or adipose tissues to regulate their metabolism and development [20]. Adipose tissue is derived from the mesoderm [21], which is divided into paraxial mesoderm and lateral mesoderm, producing white adipocytes and brown adipocytes; the latter is produced only by the paraxial mesoderm [22,23,24]. Both WAT and BAT have similar origins, but different developmental modes and morphologies (Figure 1). The progenitors of white, brown, and beige fat are mesenchymal stem cells with different molecular markers, e.g., the precursor cells of white fat are Myf5^+^ progenitors while brown fat progenitors are the opposite. In the presence of external stimuli, white adipose beige adipose can be interconverted under certain conditions. In contrast, brown fat is darker and more mitochondria-rich, beige fat is second to brown fat and white fat has the least mitochondria and is not thermogenic, as shown in the Figure 1.

White adipose tissue appears and develops later than BAT. White adipocytes are differentiated after birth, while the brown adipocytes are differentiated during the embryonic stage [25]. Unlike rodents, human infants maintain their body temperature after birth mainly through heat production in the perinephric region and in the brown adipose tissue on the sides of the spine [26]. Moreover, the morphologies and biological functions performed by the WAT and BAT also differ [23]. White adipose tissue has a single large lipid droplet and fewer blood vessels than the brown adipose tissue (Appendix A). The WATs are stored in the animal body under the skin and around the organs and store excess energy in the body as triglycerides. Brown adipocytes have many small lipid droplets and small cell areas with dense mitochondrial distribution. The UCP-1 (uncoupling protein 1) mediates the heat-generating respiration [27]. The BATs are formed at the embryonic stage and are matured at birth. After birth, the WATs increase in size and number. The body temperature is maintained primarily in a non-shivering thermogenesis after birth [28]. The BATs are widely distributed throughout the body [29], including the interscapular area, mediastinum, kidney, adrenal fat areas, para-arterial area, and neck areas. The number of brown adipocytes gradually decreases with age and are first disappeared in the interscapular area (Figure 2). The BATs exist in the deep parts of the body for a long time before disappearing into the surface area. The BAT and WAT develop from the distinct differentiation precursors.

Brown adipose tissue differentiation is closely linked to myogenic factor 5 (Myf5). The Myf5^+^ progenitor cells can be induced to differentiate into skeletal muscle cells, central rawhide sarcomere, and classic brown adipocytes [30]. White adipocytes are formed by the differentiation of the vascular and stromal layer by Myf5^−^ [31]. Therefore, the metabolic properties of BAT are more like those of the skeletal muscle cells and are primarily reflected in its structure and mitochondrial abundance in two ways. It is worth mentioning that rodents have BATs throughout their lives [32] and have become the ideal research models. In recent years, researchers have focused on the thermogenesis mechanism of BATs for anti-obesity research. Numerous studies have shown that, under certain conditions, WATs and BATs are interconvertible [33]. Nuria et al. indicated that inhibiting the expression of MKK6 (mitogen-activated protein kinase kinase 6) in WAT could increase the level of oxidative phosphorylation of UCP-1 to promote the WATB [34].

Besides the traditional WATs and BATs, there is another type of fat known as a beige fat. Cold stimulation causes the brown-like fat cells to appear in some WATs in mammals. This process is called “white fat browning (WFB)” [35] (Figure 1). These brown-like fat cells in the mammals, which were induced by specific factors, were termed “beige fats” by researchers. The origins of classic BAT and beige adipose tissue are clearly different. The BATs are derived from the Myf5^+^ cells, while the beige adipocytes are formed by the trans-differentiation of WATs [36] or derived from Pdgfra+ fat precursor cells [37]. Furthermore, the gene expression markers, such as TbxI, Tmem26, and CD137, are reported in the beige fat [38]. In contrast to brown fat, beige fat cells are close in colour to white adipose tissue and beige fat only expresses thermogenic genes upon stimulation [7]. When beige fat is stimulated by external stimuli and starts to produce heat due to the activation of thermogenic genes such as UCP-1 by cAMP signalling, the morphology of lipid droplets of beige fat changes and gradually forms multi-lamellar adipocytes, which resemble brown adipocytes in morphology [39]. The functions of beige adipose tissue are like those of the BATs, which consume energy, improve metabolism, glucose tolerance, and insulin sensitivity, and alleviate metabolic diseases caused by the excessive WAT deposition [40,41]. As a result, encouraging the WFB might be treatment option for obesity and other obesity-related metabolic diseases.

## 3. Regulators of WFB

White fat browning is a relatively a complicated process. In addition to the environmental factors, nutrients, and body metabolites, other regulatory factors are also involved in the deposition and thermogenesis of brown fat, such as PRDM16, Zfp516, PGC-1α, PPARr, C/EBP-β family members, HSF1 and IRF4 regulators [42]. The injection of PPARγ agonists in mice could significantly promote the brown fat marker genes, such as UCP-1 and PGC1a (Table 1).

The PPARγ is expressed in the mammalian adipose tissues, vascular smooth muscle tissues, and cardiac muscle tissues. The PPARγ is more abundantly expressed in WAT than the UCP-1 and is involved in regulating the proliferation and differentiation of adipose tissues [43,44]. In addition, PPARγ, in the activated state, could also lower blood glucose levels, increase insulin sensitivity, and reduce inflammatory response, thereby maintaining the normal metabolism in the body. Subsequently, studies reported that PPARγ had a positive regulatory role in inducing WFB based on its key role in regulating the proliferation and differentiation of adipose tissues and enhancing the insulin sensitivity. Both the SIRT1 and SIRT3 of the SIRT family of histone deacetylases could activate PPARγ by deacetylation; a process, which is also prone to β-adrenergic effects of norepinephrine. The factors PPARγ, PGC-1α, and PRDM16 could also play synergistic roles, acting as mutual coactivators, to promote the WFB-associated processes. Therefore, PPARγ has been recognized as a powerful regulator and agonist in WFB and its abnormal expression often leads to the different degrees of obesity syndrome. The PPARγ agonists, such as rosiglitazone, have been identified as potential browning agents and are widely used in clinical trials. Rosiglitazone belongs to the thiazolidinedione group of hypoglycaemic agents and is one of the insulin sensitizers [45], mainly activating a specific receptor in the body, which increases the sensitivity of the body to insulin, like muscle, liver and fat, in order to lower blood sugar and improve lipid metabolism [46]. Rosiglitazone also promotes the redistribution of fat in the body, causing visceral fat to shift under the skin, which may be related to the effect of increasing insulin sensitivity, as well as improving the beta-cell function of the pancreatic islets and treating fatty liver and obesity in combination with diabetes [47].

Uncoupling proteins (UCPs) are a family of transporter proteins, which are present in the inner mitochondrial membrane. Among them, UCP-1 is expressed only in the BAT and beige fat cells, and UCP-2 is expressed in many organs, while UCP-3 is expressed in the skeletal muscle cells. Because the β3-AR are found mainly in adipose tissues, their selective agonists stimulate the adipose thermogenesis as well as lipid mobilization in the WAT. Therefore, the role of UCP-1 in thermogenesis has been extensively studied. Its role is under the direct sympathetic control, exerting an uncoupled oxidative phosphorylation while increasing the ATP production in the mitochondria, which is mediated mainly through the β-adrenergic action of norepinephrine. In addition, UCP-1 also plays an important role in WFB, activating the regulation of mitochondrial function under cold stimulation or other external stimuli. The UCP-1 was significantly positively correlated with the mitochondrial content and respiration and upregulated the expression of Parkin-dependent and independent mitochondrial autophagy-associated marker genes. It could also regulate the expression levels of endolipin, lipocalin Cidea, and Nrg4, thereby showing a guiding and central role in the overall WFB and thermogenesis. Given these reasons, UCP-1 has been widely studied as a marker of BAT and WAT browning (As shown in Figure 3).

The SIRT1 protein, a member of the SIRT family, mainly regulates the post-transcriptional modifications and belongs to a family of histone deacetylases. In bacteria, yeasts, and mammalian cells, the SIRT family of proteins have a highly conserved similar catalytic core region. The SIRT1 is mainly localized in the nucleus. However, it can be induced by UCP-1 to migrate into the cytoplasm under cold stimulation, where it plays a regulatory role mainly through the lys268 and lys293 pathways. It can also deacetylate PPARγ to enhance the WFB activity and inhibit lipogenesis in beige adipose tissue as well as stimulate lipolysis. In addition, SIRT1 can also regulate the deacetylation of various proteins through metabolic pathways, such as AMPK, FOXO, and mTOR, which have indirect regulatory effects. Therefore, researchers refer to the correlations among SIRT1, PPARγ, PGC-1α, and AMPK as the SIRT1/PPARγ/PGC-1α and AMPK/SIRT1/PGC-1α pathways.

The PGC-1α is a transcriptional co-activator, mediating many biological responses related to energy metabolism [33]. In the past, PGC-1α was considered a powerful transcriptional co-activator of PPARγ and was also named PPARγ coactivator 1α due to its initial role as a PPARγ- interacting protein. However, subsequent studies revealed that PGC-1α could bind to a variety of transcription factors [48], including PPARγ, NRF, and MEF2C, thereby participating in the mitochondrial biosynthesis, adaptive thermogenesis, skeletal muscle type switching, fatty acid oxidation, oxidative stress, gluconeogenesis, and other metabolic processes. In general, the PGC-1α expression is low in WAT. However, in response to external stimuli, such as cold, the activation of sympathetic nerves and β-adrenaline, as well as the ectopic expression of PGC-1α, can significantly increase the expression of UCP-1 and other brown-specific genes in WAT, thereby directly or indirectly regulating the mitochondrial respiration and biosynthetic capacity as well as WFB. The extensive binding capacity of PGC-1α is indispensable for the promoting the WFB and its thermogenesis. Therefore, it has a high potential for clinical studies to be used as therapeutic target for drugs and agonists.

The regulatory factor PRDM16 is abundantly present in BAT and browning WAT [49,50], regulating the function of classical brown adipocytes and beige adipocytes. Numerous previous studies have reported the importance of PRDM16 in converting the WAT to beige adipocytes. The deletion of PRDM16 could induce muscle differentiation by decreasing the thermal gene expression and increasing the muscle-specific gene expression in BATs. Furthermore, PRDM16 could induce an almost complete brown fat genetic program, including the increased mitochondrial biogenesis, cellular respiration, and selective expression of brown fat-associated genes. The highly expressed PRDM16 can form a complex with zinc finger proteins and bind to the promoter region of UCP-1, thereby upregulating the UCP-1 transcription. The activated UCP-1 can block the phosphorylation process during ATP production. Then, the potential energy of the proton gradient no longer flows back down the ATP synthase but follows the UCP-1 uncoupling channel back to the mitochondrial matrix, releasing potential energy as heat energy [50,51]. In addition, PRDM16 can also interact with transcription factors, such as PPARα, PPARγ, and C/EBP family members, and cofactors, such as PGC-1α, and enhance their transcriptional activity, thereby playing an indirect role in the WFB.

White fat browning is a complex process, which is regulated by multiple nutrients and metabolites and affects the related signalling pathways. White fat browning is regulated by numerous regulatory factors, such as UCP-1, PPARγ, PRDM-16, and PGC-1α [52]. The classical theory of WFB is the excitation of sympathetic nervous system and activation of the β3-ARs signalling pathway induced by PRDM-16 and SIRT1, which exert deacetylation effects on PPARγ [53]. The latter process is dependent on the induction and translocation of the transcriptional co-activator PGC1α into the nucleus and its association with the gene promoter region. In addition to sympathetic nerves, cellular energy-sensing is a driving force in regulating browning by regulatory transcriptional networks, such as AMPK, which inhibits fat deposition and induces WFB by downregulating the ACC expression and inhibiting the gluconeogenesis, among other pathways [54]. Therefore, it is theoretically possible to treat a variety of obesity and other metabolic disorders by inducing and activating the production of beige fat to regulate the expression of thermogenesis-related genes.

## 4. Inducing Factors of White Fat browning

### 4.1. Cold Stimulation

To date, numerous studies on the activation mechanism of BAT have been published. Under cold stimulation, the transcription factor proliferator-activated receptor (PPARγ), peroxisome proliferator-activated receptor-gamma coactivator -1α (PGC-1α) can activate BAT to produce heat [55]. In the meantime, free fatty acids, which are hydrolysed in the lipid droplets of brown adipocytes in response to cold stimulation can not only promote the division of mitochondria but can also increase the uncoupling activity of mitochondria, leading to an increase in the thermogenic properties of BAT [56]. The BAT is an important organ for heat production and hydrolyses triglycerides in brown adipocytes mainly by β-oxidation in the mitochondria. As a result, the lipid droplets and mitochondria are necessary for BAT thermogenesis. Cold stimulation can activate BAT by releasing adrenaline through sympathetic nerves, increasing the thermogenesis and expression of lipid metabolism-related genes and promoting the subcutaneous accumulation of beige fat [57]. The effects of cold stimulation on the body have been reported in mice and humans [58]. Mice stimulated with cold at 10 °C for 2–6 h showed an increase in the scapular brown fat oxidation metabolism and body surface temperature, as well as the expression levels of mitochondrial and thermogenesis-related genes [59]. Moreover, after a healthy adult was exposed to 19 °C for 2 h, FDG-PET/CT (Fluorodeoxyglucose-Positron Emission Tomography/Computed Tomography) showed that FDG in the supraclavicular and paraspinal adipose tissue was significantly absorbed in adipose tissue in a specific area, representing BAT. Metabolic activity suggested that cold stimulation could induce and activate the adipose tissue browning, and the cold-induced absorption of supraclavicular fat FDG was significantly negatively correlated with BMI and visceral fat [60]. In addition, the cold or β-adrenoceptor agonists could stimulate the production of heat in humans and mice [61], and at the same time, induce FGF21 expression in adipose tissues. Fibroblast growth factor 21 (FGF21) regulates protein levels of PGC-1α at the post-transcriptional level and activates subcutaneous fat browning as well as thermogenesis of brown fat in mice [62]. FGF21-knockout mice showed impaired intolerance to cold and WFB [63]. By activating the β3 adrenergic receptor-cyclic adenylate-protein kinase, cold stimulation could activate a signalling pathway, which promoted the lipolysis of stored triglycerides to release free fatty acids and increase expression levels of thermogenesis-related genes [64]. A recent study reported that histone deacetylase 1 (HDACl) was a negative regulator of the brown fat thermogenesis. Cold exposure and β-adrenergic receptor activation could activate HDACl and brown fat-specific genes. In order to promote the expression of brown fat-specific genes, the subregions were separated [65]. Other regulatory factors are also involved in the adaptive thermogenesis induced by cold stimulation. Janus kinase2 (JAK2) is a signal transduction factor for various hormones and growth factors, regulating the development and function of fat. The JAK2 and UCP-1 were both elevated in cold-stimulated brown fat and the JAK2-knockout mice showed lower levels of brown fat marker genes and a lower core body temperature, indicating that this protein could also be an important part of the brown fat adaptive thermogenesis [66]. The transcription factors and their cofactors, including ATF2 (activating transcription factor) [67], Zfp516 (zinc finger protein 516) [68], and PGC-1α [62], are also involved in the regulation of cold-stimulated adaptive thermogenesis of adipose tissues.

### 4.2. Exercise

Exercise can increase the number of brown fat progenitor cells and the expression level of the brown fat marker gene UCP-1. Simultaneously, exercise can also promote WFB [69], as evidenced by an increase in the number of white fat mitochondria and upregulation of UCP-1, PGC-1α, and Dio2 (Deiodinase Iodothyronine Type II) and other brown fat marker genes [70]. Exercise could promote the PGC-1α expression in muscles in both mice and humans, which can regulate the brown fat UCP-1 expression level and thermogenesis. As a result, the mice, overexpressing PGC-1α in skeletal muscle, have been frequently used as exercise models [33]. Exercise increases the expression of brown fat marker genes UCP-1 and CIDEA (cell death-inducing DNA fragmentation factor, alpha) in inguinal fat, indicating that it browns subcutaneous fat in mice.

In contrast, numerous studies have reported that exercise training in mice could promote the expression of brown fat-specific genes, such as UCP-1 in visceral fat but showed no effect on the subcutaneous fat. In addition, exercise induced WFB in mice was also affected by diet. The subcutaneous inguinal fat in the exercise mice showed a browning phenotype. The proteins involved in the thermogenesis and oxidation of adipose tissue, including PGC-1α, UCP-l, adipose triglyceride lipase (ATGL), adenosine monophosphate-activated protein kinase (AMPK) as well as increased phosphorylation and palmitate oxidation levels were observed. However, the exercise-induced browning phenotype decreased significantly in the high-fat diet (HFD)-fed mice [71,72].

In recent years researchers have found that High Intensity Interval Exercise (HIIT) has similar or better effects than continuous aerobic exercise in relieving symptoms of insulin resistance [73]. There is no clear definition of HIIT at this stage, but it generally consists of multiple repetitions of short or prolonged exercise at near or extreme intensity (85–95% VO2max or HRmax), interspersed with intervals ranging from a few tens of seconds to a few minutes [74]. Studies have shown that HIIT exercise can improve metabolic disorders in rodents, mainly by improving biological processes such as transmembrane transport of fatty acids and β-oxidation, with HIIT being significantly more effective than other forms of exercise [75]. Interestingly, HIIT also significantly promotes the expression of genes associated with white adipose browning in subcutaneous adipose tissue in mice, such as UCP-1, facilitating the transition from white to beige adipose tissue [76]. However, in a 12 week training study (30 min at 70% of maximal power output 2 times/wk) in obese or sedentary humans, this high intensity training and resistance training did not alter the expression of brown fat markers such as UCP-1 and Prdm16 in the adipose tissue of obese individuals [77]. However, there is controversy as to whether exercise at different intensities promotes beige-coloured white adiposity in humans. A study by Berenice et al. [78] showed that a 12 week bicycle-training program (3 times per week, intensity 70–80% HRmax) in people with different BMIs significantly promoted abdominal fat in the exercise group significantly promoted the expression of beige fat marker genes for abdominal fat in the exercise group. However, the results of Tsiloulis’ study showed that long-term endurance exercise training did not promote beige or brown abdominal fat in obese men [79]. The interspecies differences in the effects of exercise on adipose tissue browning in humans and rodents may be due to different patterns of adipose tissue gene expression between species, and the authors suggest that the small sample size of studies conducted on white fat browning in humans, the difficulty of collecting comprehensive subcutaneous adipose tissue from subjects, and the incomplete mapping of exercise intensity and duration of exercise have contributed to these interspecies differences. The results vary between species. Moreover, irisin plays an important role as a signalling molecule in the regulation of white adipose tissue browning during exercise, as described in detail in Section 4.4.8.

### 4.3. Nutrients

Brown fat plays an important biological role in the regulation of energy metabolism throughout the body, in addition to a similar role for white fat browning (beige fat). Therefore, activation of brown fat as well as beige fat thermogenesis is an excellent target for improving the body’s energy metabolic balance and promoting weight loss. In this paper, we summarise data on small molecules of food origin that can modulate energy metabolism in obese patients by activating brown or beige fats. Furthermore, studies on mostly food-derived small molecules have only been conducted at the mouse level and have not been carried out in human populations for the time being. Therefore, further studies on the role of these small molecules in the regulation of beige fat and brown fat thermogenesis in humans are still needed.

#### 4.3.1. Sesamol

Sesamol is a fat-soluble lignan compound, which is the main component of the sesame oil fragrance. In addition, sesamol is also a critical quality stabilizer for sesame oil with a wide range of biological activities, including anti-oxidation and scavenging of free radicals. Sesamol inhibits the white adipogenic genes, such as the mRNA levels of PPARγ, acetyl-coenzyme A carboxylase (ACACA), steroid regulatory element-binding protein 1c (SREBP-1c), and fatty acid synthase gene (FASN). It can also inhibit the increased number of lipid droplets in BAT and promote the expression of brown fat marker genes, including UCP-1, FGF21, and COXII [80]. Moreover, in-vitro studies reported that treating the 3T3-L1 model cells with a specific dose of sesamol could significantly inhibit the accumulation of lipid droplets in adipocytes [81].

#### 4.3.2. Resveratrol

Resveratrol (RSV) is an anthraquinone terpenoid. It was first discovered in the rhizomes of resveratrol by Japanese researchers in 1939 [82].

Resveratrol is widely distributed and biosynthesized by the plants in the free state as well as glycoside-binding state. It can activate Silent Information Regulator 1 (SIRT1) to play a similar role in the calorie restriction (CR) in various functions, such as the regulation of organism’s lifespan and inhibition of premature cell aging. Resveratrol inhibits the lipid peroxidation and lipoprotein modification caused by the reactive oxygen species (ROS) and affects the essential fatty acid metabolism.

Resveratrol is an agonist of SIRT1, increasing the cell viability and activity of the SIRT1 enzyme as well as their affinity for SIRT1-acetylated substrates [83]. Shi et al. [84] reported that the expression of SIRT1 was significantly higher in the WAT than that in the BAT and RSV could reduce the lipid accumulation by increasing the deacetylation of PPARr through SIRT1. Moreover, activated SIRT1 could deacetylate PGC1α to achieve the lipid glucose-lowering effect [85]. In addition, pterostilbene could also promote the WFB. In addition, the results of the study by Suo et al. [86] showed that long-term feeding of resveratrol to db/db mice significantly promoted lithocholic acid (LCA) in the plasma and faeces, thus indirectly promoting browning of white fat.

Furthermore, grape pomace extract (GPE), which is functionally like RSV is extracted from grape pomace as a residue in the winemaking process. The GPE is mainly composed of berry skins and seeds, containing a lot of phenolic compounds. The most abundant polyphenols, which are found in Malbec grape pomace, include flavanols (catechins, epicatechins, and proanthocyanidins), flavonols (quercetin), stilbene (RSV), and anthocyanins [87].

The GPE prevents the palmitate-mediated down-regulation of FNDC5/irisin protein expression and secretion by activating the PGC-1α. Circulating irisin can upregulate the expression of UCP-1 in WAT and promote the formation of brown-like adipocytes [88].

#### 4.3.3. Ellagic Acid

Ellagic acid is a phytochemical abundant in fruits and vegetables, in particular berries, with reported pharmacological effects, such as anti-oxidation and anti-cancer. In order to maintain cell viability, ellagic acid could inhibit the protein expression of cyclin A, thereby inhibiting the formation of fat in the 3T3-LI model adipocytes [89]. Ellagic acid could inhibit the accumulation of triglycerides in adipocytes by downregulating the expression of the key genes, such as PPARγ, and the transcription factor CCAAT-enhancer-binding protein α (C/EBPα). A study by Ning Chao [90] reported that the ellagic acid could inhibit the HFD-induced accumulation of lipids in rats as well as could down-regulate the expression levels of lipid synthesis-related proteins, such as PPARγ, C/EBPβ, and C/EBPα, and upregulate the BAT marker proteins UCP-1 and UCP-1 in WAT. The PGC-1a can inhibit the transformation of pre-adipocytes into mature adipocytes, thereby promoting the WFB. Yu et al. [91] reported that the pomegranate ellagic acid could promote lipid metabolism, reduce intracellular lipid levels, and inhibit the activities of lipoprotein lipase (LPL) and glucose-3-phosphate dehydrogenase (GPDH) thereby inhibiting the formation of fat in adipocytes, upregulating the PPARγ and adipocyte fatty acid-binding protein (aP2) levels, and down-regulating the obesity gene ob [90]. A study by Liu [92] suggested that the pomegranate ellagic acid could affect lipid metabolism mainly by inhibiting the gene expression of SREBP-1c and FASN, while promoting that of LPL, thereby inhibiting the lipid accumulation.

#### 4.3.4. Flavan-3-Alcohol

Flavan-3-ols are polyphenols found in many plant foods, such as cocoa beans, red wine, and apples, and are considered a mixture of catechins and B-type proanthocyanidins [93]. Previous studies reported that flavan-3-ols could play an essential role in the treatments of diabetes, cardiovascular disease, tissue damage, and inflammation, indicating their extensive role in the metabolism-related processes in the body and the maintenance of body homeostasis. Subsequent studies also reported their role in fat metabolism [93]. The treatment with flavan-3-ol could increase the sympathetic nerve activity, adrenaline contents, and the mRNA expression levels of UCP-1 in mice, suggesting that flavan-3-ol had a positive effect on lipid metabolism in mice, thereby affecting the proliferation and differentiation of brown adipocytes and related regulatory processes of WFB. Moreover, flavan-3-ols could also increase the concentration of catecholamines in plasma, which induce WFB by stimulating the activation of β-adrenergic receptors and mediating the protein kinase A signal transduction, thereby promoting the fat metabolism-related processes [94].

#### 4.3.5. Epicatechin

Catechin is a type of phenolic active substance extracted from natural plants, such as tea, and has various pharmacological activities, such as anti-tumour, anti-oxidation, and anti-bacterial activities as well as affects the heart and brain organs [95]. Epicatechin can induce WFB by promoting mitochondrial biogenesis, enhancing the mitochondrial structural and functional indicators, increasing the fatty acid metabolism, and upregulating the expression of BAT-associated genes. It can increase the expression levels of mitochondrial biogenesis-related proteins, such as PGC1α, mitochondrial transcription factor A (TFAM), SIRT1, SIRT3, and uncoupling protein 1 (UCP-1). Moreover, epicatechin can also activate the AMPK while enhancing the phosphorylation of ACC and inhibiting the expression of gluconeogenesis-related genes, which results in decreasing the fatty acid uptake and triglyceride synthesis, thereby inhibiting the lipogenesis, and increasing fatty acid oxidation. Epicatechin can activate the browning of adipose tissue and increase the cold-induced heat production in human body. It positively affects the browning of fat cells and WATs [96]. Therefore, this phytochemical has become a potential candidate for combating obesity.

#### 4.3.6. Capsaicin

Capsaicin is an alkaloid derived from pepper, which can bind to the vanilloid receptor subtype 1 (VR1) of sensory neurons in the mammals and produce a burning sensation. Capsaicin has been reported to lower blood pressure and cholesterol levels, prevent heart diseases, promote muscle growth, inhibit muscle atrophy, and can widely be used in food additives, medicine, and health care [97]. Capsaicin induces WATB and fights obesity by activating the TRPV1 (transient receptor potential cation channel, subfamily V, member 1) channel-dependent mechanism in obesity treatment [98]. Specifically, the TRPV1 channel-dependent increase in the intracellular Ca^2+^ and phosphorylation of Ca^2+^/calmodulin-activated protein kinase II and AMP-activated kinase promote the expression and activity of SIRT1, thereby triggering the WFB. Capsaicin can regulate the PPAR transcription, upregulate the specific BAT- and WAT-associated gene, stimulate SIRT1-dependent deacetylation of PPARγ and transcription factor Prdm16, and promote the interaction between PPARγ and Prdm16, thereby leading to WFB.

#### 4.3.7. Curcumin

Curcumin is a diketone compound extracted from the rhizomes of the families Zingiberaceae and Araceae. It has a wide range of biological activities, including anti-inflammatory, antioxidant, lipid-regulating, anti-viral, anti-infective, anti-tumour, and anti-coagulation activities. It has also a wide range of pharmacological activities, such as anti-liver fibrosis and anti-atherosclerosis [99]. In the fat-related processes, curcumin could also reduce body weight and fat mass, by promoting the production of beige fat cells and expression of heat-producing genes and increasing the mitochondrial biogenesis, without affecting the food intake in mice. The cold tolerance studies in rats [100] reported that curcumin could promote the expression of β3AR gene in the inguinal white adipocytes, thereby increasing the norepinephrine level in plasma and inducing the WFB [101]. In addition, curcumin could also induce the polarization of M2 macrophages by secreting interleukins IL-4 and IL-13, which could cause WFB [102].

#### 4.3.8. Berberine

Berberine, also known as berberine, is a quaternary ammonium alkaloid, which is isolated from the traditional Chinese medicine Coptis Rhizoma. It is the main effective component in the anti-bacterial activity of *Coptis*. As one of the effective anti-bacterial and antitoxic substances, it has a widespread clinical use [103]. Berberine has metabolic regulatory roles as well, such as a role in the anti-oxidation, anti-inflammatory, anti-tumour, anti-bacterial, liver protection, neuroprotection, hypolipidemic, and hypoglycaemic pathways [104]. Studies generally believe that Berberine could increase the energy expenditure in obese db/db mice, limit their weight gain, and improve insulin sensitivity, cold tolerance, and the activity of BAT. Berberine increases the expression of thermogenic genes, including UCP-1, the WAT, BAT, and primary adipocytes, through the mechanisms, involving AMPK and PGC-1α. In order to induce the brown/beige lipogenesis, berberine could increase the transcription of PRDM16 by increasing the demethylation of the active promoter region of PRDM16 gene, which is a main regulator in the browning process [105]. In summary, berberine could enhance the thermogenesis of brown adipocytes and WFB to promote the metabolic process of adipose tissue through the AMPK pathway, thereby showing its potential therapeutic significance for the treatment of obesity.

#### 4.3.9. Quercetin

Quercetin is widely distributed in the stem bark, flowers, leaves, buds, seeds, and fruits of many plants and is mostly present in the form of glycosides [106]. It has a wide range of biological activities, such as tissue anti-oxidation [107], anti-cancer [108], anti-bacterial, anti-inflammatory [109], anti-allergic, and antidiabetic activities [110], as well as a strong biological and therapeutic effect on cardiovascular diseases [111]. In order to the improve obesity, studies have reported that quercetin could reduce lipid deposition in the liver and the storage of WAT in the HFD-fed mice, thereby lowering the triglycerides contents in plasma and reducing their weight [112]. In the process of WFB, quercetin increased the expression levels of UCP-1 and brown fat marker fatty acid elongase 3 (Elovl3). The specific mechanism involves the quercetin/AMPK/SIRT1/PGC1α pathway, directly acting on the adipocytes and increasing the mitochondrial biogenesis to induce browning [113]. Interestingly, quercetin is also known as phytoestrogens due to its ability to bind to the oestrogen receptor (ER). Studies have shown that the mouse ER-β ligand LY3201 could induce WFB by directly regulating the sympathetic ganglia and adipocytes, increasing oxygen consumption, and reducing the body weight [114]. Therefore, the effect of quercetin to induce browning in humans and improve the triglyceride metabolism should be further investigated.

#### 4.3.10. Fucoxanthin

Fucoxanthin is a natural pigment of xanthophylls in carotenoids in brown algae, diatoms, golden algae, and yellow-green algae. It has anti-tumour, anti-inflammatory, antioxidant, and anti-obesity effects as well as protective effects in nerve cells. It is widely used in the medicines and skincare, beauty products, and health care products [115]. Fucoxanthin can improve insulin resistance and lower blood sugar levels [116,117]. In a subsequent experiment, an increase in the intake of fucoxanthin in mice increased the mRNA expression levels of β3-AR in WAT, which might stimulate the sympathetic nervous system and upregulate the expression of UCP-1, thereby generating heat [118]. Studies have found that, in rodents, the anti-obesity effects of fucoxanthin were related to the activation of brown fat and WFB. However, fucoxanthin and its metabolite fucoxanthin neither induced the browning of human adipocytes nor changed the mRNA expression levels of PGC-1α, PPARα, PPARγ, PDK4, and FAS [119]. Therefore, the clinical use of fucoxanthin as a drug to treat obesity requires further studies.

#### 4.3.11. Menthol

This is a terpenoid organic compound, which is extracted from the leaves and stems of peppermint [120]. It is used as a stimulant in medicine and acts on the skin or mucous membranes, showing cooling effects and relieving itching. It can be taken orally as an expel medicine for headaches or the inflammation of nose, pharynx, and throat. It can also be used as a flavouring agent in toothpaste, perfume, beverages, and candies. Studies have shown that inducing the transient receptor potential cation channel, subfamily M, member 8 (TRPM8) activation by dietary menthol might enhance the WFB and improve the diet-induced obesity. The TRPM8 is an ion channel, which can detect cold stimuli in the thermal sensory system. Its activation can upregulate the expression levels of UCP-1 and PGC-1α, thereby enhancing the lipid metabolism. Menthol could also increase the levels of iron and copper in adipose tissue by activating the TRPM8. The concentration of related metals in adipose tissues is closely related to the health and differentiation of adipose tissues and WFB [121]. Iron and copper are the essential components of the inner membrane complex of mitochondria, which constitute the electron transport chain. Therefore, these metals might participate in the energy metabolism by playing a role in mitochondria and BAT activation. In summary, TRPM8 might participate in WFB by increasing the expression levels of thermogenesis- and metabolism-related genes. As an activator of TRPM8, menthol might be a promising candidate for the treatment of obesity and other metabolic diseases.

#### 4.3.12. Chlorogenic Acid

Chlorogenic acid, a common phenolic acid, is found in fruits, vegetables, and traditional Chinese medicines and is responsible for a variety of physiological activities. It is also one of the main anti-bacterial and anti-viral pharmacological components of the honeysuckle. Although it has a wide range of anti-bacterial activities, it causes sensitivities among some people [122]. Chlorogenic acid could attenuate the high-carbohydrate and HFD-induced cardiovascular and liver metabolic disorders in rats caused by showing anti-obesity effects. It could also stimulate the production of brown fat cells by promoting the glucose uptake and mitochondrial function [123]. Furthermore, to induce WFB, chlorogenic acid and caffeic acid could act synergistically and activate the browning program in human adipocytes by upregulating the expression of AMPK and other browning-associated genes at the transcription and protein levels. Chlorogenic acid could exert anti-diabetic and anti-obesity effects through the AMPK pathway, enhance the expression levels of PPARγ, PRDM16, and PGC-1α in BATs and WATs, increase the insulin production, and inhibit key enzymes in lipid biosynthesis [124].

#### 4.3.13. Chrysin

Chrysin is a flavonoid compound, which is extracted from plants in the Bignoniaceae family, which has anti-oxidation, anti-tumour, anti-cancer, anti-viral, anti-hypertensive, anti-diabetic, anti-bacterial, anti-allergic, and other pharmacological and physiological activities. Moreover, these compounds are widely distributed in plants and have relatively low toxicity and high biomedical research potential [125]. The WFB potential of chrysin has attracted great attention. Chrysin can regulate fat cells by reducing adipogenesis, increasing fat oxidation, and inducing browning [126]. Activation of AMPK in 3T3-L1 cells by increasing the expression of p-AMPK, chrysin could significantly upregulate the expression of PGC-1α, UCP-1, PRDM16, PPAR family proteins, and other browning proteins in the 3T3 model cells, thereby promoting the lipolysis, fat oxidation, and thermogenesis and reducing the fat production. Chrysin can also stimulate the expression of perilipin (PLIN) which inhibition of lipolysis in the presence of PKA stimulation and enhancement of lipolysis in the presence of PKA stimulation. A study reported that the increased expression of PLIN was a cause of the anti-obesity effects of chrysin and recruitment of brown adipocytes to the white adipocytes [127]. The in vitro data obtained from the 3T3-L1 adipocytes showed that chrysin could significantly induce the fat browning. However, its role in the body remains to be explored.

#### 4.3.14. Cinnamic Aldehyde

Cinnamic aldehyde is an organic aldehyde compound, which is abundantly present in the Sri Lankan cinnamon oil, ageratum oil, hyacinth oil, rose oil, and other cinnamon plants [128]. Cinnamic aldehyde induced the browning of WATs in the HFD-fed mice and showed therapeutic efficacy in obesity. The cinnamic aldehyde treatment specifically reduced the body weight, fat mass, food intake, serum lipid, free fatty acids, and leptin levels [129]. It also improved insulin sensitivity in the HFD-induced obese mice, thereby reducing the insulin resistance in obese mice [130]. In addition, it can also inhibit the hypertrophy of adipose tissues and induce the browning of WATs and UCP-1 expression. Furthermore, it could enhance the expression of PPARγ, PRDM16, and PGC-1α proteins in the WATs and BATs, increase the mitochondrial respiration and enhance lipid metabolism [131]. Cinnamic aldehyde might play a role in the treatment of obesity and other related diseases in the future, but its potential mechanism is needed to be investigated.

#### 4.3.15. Luteolin

Luteolin is a natural flavonoid compound, which is abundantly present in various edible and medicinal plants, such as pepper, celery, thyme, mint, honeysuckle, etc. A recent study showed that dietary luteolin improved the diet-induced obesity and insulin resistance in mice [132].

Luteolin promoted the browning of differentiated primary brown cells and subcutaneous adipocytes by regulating the AMPK/PGC1α pathway. The luteolin treatment increased the protein levels of UCP-1, PGC1α, and SIRT1 and the phosphorylation levels of AMPKα and ACC. In addition, luteolin could also induce the beige cell-specific markers in the differentiated primary subcutaneous adipocytes [133]. At present, studies have reported that dietary luteolin could enhance the activity of brown adipocytes, formation of beige adipocytes and related thermogenesis processes, and improve the diet-induced obesity and insulin resistance [133].

#### 4.3.16. Taurine

Taurine is a sulphur-containing amino acid in animals (a nonprotein amino acid), which is synthesized from methionine and cysteine. Studies have shown that taurine can participate in various biological and physiological functions, including membrane stabilization, salt coupling, immune regulation, and anti-oxidation [134]. In a recent study, taurine treatment could induce the browning of WAT, which depended on the mRNA induction of PGC1 in WATs mediated by the AMPK signal. The results showed that taurine could play an essential role in regulating the production of adipose tissues and plasticity of white fat on BAT and provide a mechanism for the protective effects of taurine on obesity [135].

#### 4.3.17. Emodin

Emodin is a natural anthraquinone derivative and has various pharmacological effects, including the lowering of blood lipids and regulation of glucose utilization (in the rodent model species) [136]. Emodin significantly increased the mRNA expression levels of beige adipocyte markers, such as Cd137, Tmem26, and Tbx1 in inguinal white adipose tissue (iWAT), and UCP-1, CD36, FABP4, and PPARα as well as the expression level of inhibitory protein expression in both the iWAT and BAT. Studies reported that emodin could improve the obesity and metabolic disorders in obese mice. In addition, it could also promote browning in the iWAT and activate BAT activity [137]. In addition, the changes in lipid contents in the iWAT and BAT caused by emodin treatment were highly specific to certain molecular lipid species, indicating that the changes in the tissue lipid contents reflected the selective remodelling of iWAT and BAT by glycerophospholipids and sphingolipids in response to the emodin treatment.

#### 4.3.18. 3′-Hydroxydaidzein

3′-Hydroxydaidzein (OHD) is a daidzein metabolite (DAI), which is present in the fermented soy products, such as miso. The DAIs have been reported to affect the lipid accumulation, but the effects of OHD on lipid accumulation require further investigations. A study investigated the effects of OHD on the HFD-induced obese. The results showed that as compared to the HFD group, the mice treated with 0.1% OHD (HOHD) showed significant reduction in their body weight and groin fat without changing their food intake. Hyperlipidaemia in the HOHD and DAI groups was relieved by lowering the serum levels of triglyceride and total cholesterol. As compared to the HFD group, the HOHD and DAI groups showed significantly smaller size of the fat cells in the inguinal as well as increased expression levels of the PRDM16, C/EBPβ, p-p38, SIRT1, PGC1α, and UCP-1 proteins. In addition, the gut microbiota of the mice was enriched with Lachnospira and GCA_900066225 in the OHD and DAI groups as compared to the HFD group. In conclusion, OHD could improve the HFD-induced obesity in mice by stimulating the browning of WAT and regulating the gut microbiota [138].

#### 4.3.19. Rice Bran

Rice bran is a nutrient-rich and resource-rich by-product, which is produced during the processing of rice grains and accounts for about 10% of the rice weight. It has vital health-promoting effects, including anti-cancer, anti-obesity, anti-diabetics, anti-dyslipidaemia, and anti-inflammatory activities [139]. The oral administration of rice bran could also significantly upregulate the expression level of UCP-1 protein and coding genes and downregulate those of TCF21 and HOXC8 (WAT-specific proteins). In addition, RRB IRB and is also effectively increased PRDM16 and PGC-1α expression [140].

#### 4.3.20. Purple Sweet Potato (PSP)

Purple Sweet Potato is a functional food rich in anthocyanins, having various potential biological and pharmacological effects. As compared to the HFD control mice, the PSP-treated mice showed a significant upregulation of browning-related gene expression, including PGC1α and UCP-1, in the iWAT. Similarly, in the mouse adipocytes treated with PSP, the protein levels of PGC1a and UCP-1 increased. These results indicated that PSP might regulate the energy expenditure by regulating these molecules promotes browning of white adipose tissue and prevent the HFD-induced metabolic abnormalities [141].

In addition to the above nutrients, numerous studies have shown that the dietary apple polyphenols [142], strawberry methanol extract [143], dietary silk peptides [144], *Lactobacillus amylovorus* KU4 [145], chitosan and chito-oligosaccharides [146], Sargassum [147], freeze-dried *Aristotelia chilensis* berries [148], cardamom [149], psoralen seeds prenylated flavonoid standardized extract [150], fermented *Cordyceps militaris* extract [151], genistein [152], broccoli [153], and allicin [154] could also induce the browning of WATs.

The above molecules of dietary origin may be one of the dietary components of obese individuals with disorders of glucolipid metabolism that reverse obesity by stimulating thermogenesis in brown or beige adipose tissue. In this section, we also summarise the molecular mechanisms regarding the contribution of various molecules of dietary origin to the stimulation of WFB. First, capsaicin [98], epicatechin [96] and quercetin [110] promote WFB by activating the cAMP signalling pathway. Curcumin [102] and fucoxanthin [119] promote white adipose tissue browning by enhancing the expression of the β3AR gene in white adipose tissue, thereby mimicking the effect of cold stimulation. In addition, the secretion of cytokines that promote thermogenesis in brown fat is also an important way for dietary molecules to regulate white adipose tissue browning, such as chlorogenic acid [122], chrysin [127], lignan [155] and taurine [135]. The above nutrients suggest to researchers that dietary modulation of the metabolic syndrome in obese individuals may become one of the future treatments for obesity. However, relying solely on the findings of the current phase of research is not sufficient. Many of these nutrients have been studied at the in vitro level, with studies using mouse 3T3-L1 as a model, but the changes in gene expression data at the cellular level are not sufficient to demonstrate similar results in human cell lines or even in humans. Furthermore, such studies of white fat browning at the cellular level do not demonstrate whether a nutrient is toxic to other tissues and organs in mice or humans. In addition, some nutrients have been tested for in vivo functional validation in rodents. However, in the case of resveratrol, for example, it is high doses of resveratrol that promote white fat browning. However, long-term high dose administration of resveratrol is not feasible for humans. Furthermore, because the AMPK signalling pathway is involved in a very wide range of biological processes in the human body, it is impractical to use AMPK agonists in humans to activate brown, beige adipose tissue for thermogenesis. More importantly, subcutaneous fat as well as visceral fat is more susceptible to browning in rodents than in humans. Therefore, more needs to be reported on the molecular mechanisms and safety of dietary molecules for the treatment of obesity metabolic syndrome through a particular dietary molecule or dietary combination as a dietary therapy.

### 4.4. Signal Molecule

#### 4.4.1. Glucocorticoids (GCs)

Glucocorticoids are important corticosteroid hormones, regulating the body’s metabolism. Cortisol promotes lipolysis, facilitates gluconeogenesis during exercise or during biological processes with high energy expenditure, and raises blood glucose to reduce insulin sensitivity of peripheral tissues [156]. However, excess cortisol can lead to centripetal obesity in humans and the transitory accumulation of visceral fat triggering diseases such as type 2 diabetes and Cushing’s syndrome [157].

Studies reported that GCs could significantly downregulate the expression level of UCP-1 in BATs. However, the administration of a GCs inhibitor, RU486 could significantly increase the expression levels of BAT-associated functional genes, such as UCP-1 [158]. In in vivo, in vitro, and viral infection studies, GCs could inhibit the BAT-associated functional proteins, including UCP-1 and PR domain protein 16 (PRDM16), in mouse WATs and suppress the oxygen consumption. The GCs regulated the expression of miR-27b by binding to the glucocorticoid response element (GRE)-binding region in the upstream promoter of miR-27b. The miR-27b can inhibit the mRNA of PRDM16 target gene through the predicted auction site in the 3′-UTR of the gene. The expression of PRDM16 can reduce the WFB. Therefore, this might be a possible molecular mechanism for the inhibitory effects of GCs on WFB. The miR-27b can inhibit the differentiation and maturation of WATs by acting on the target PPARγ gene in human adipocytes [159].

#### 4.4.2. Neuregulin 4 (NRG4)

Neuregulin has four different types, belonging to the epidermal growth factor (EGF) family of extracellular ligands, which play an essential role in the regulation of cellular growth and tumour formation [160,161]. Neuregulin 4 is mainly expressed in and secreted by BATs and is hydrolysed by a protease, producing extracellular fragments. These extracellular fragments have EGF-like domains and can reach target cells through blood circulatory system in an autocrine, paracrine, or endocrine manner, thereby affecting the target cells [162]. NRG4-knockout mice experiments showed that NRG4 was secreted by the adipocytes and directly affected the target tissue liver. The NRG4 can inhibit lipid synthesis and metabolism in hepatocytes. The NRG4 was produced in the BATs under the combined action of protease and cleavage of extracellular active protein fragments. The extracellular fragment binds to the v-erb-b2 avian erythroblastic leukaemia viral oncogene homolog 4 receptor on the surface of hepatocytes. Nrg4 signalling in hepatocytes resulted in cell-autonomous inhibition of the SREBP1c/lipogenic pathway through trans-repression of LXR by STAT5 [163], the inhibition of liver X receptor (LXR) activity activated the intracellular transcription and also activated and phosphorylated the signal transducer and activator of transcription 5 (STAT5), while significantly reducing the expression level of SREBP-1c. The changes in these factors could decrease the mRNA expression levels of acetyl-CoA carboxylase (ACC), stearoyl-coenzyme A desaturase 1 (SCD1), and FASN. These changes could also inhibit the lipid synthesis in liver cells. However, NRG4 affects the lipid output, intake, and oxidation, thereby showing evident effect [164]. The NRG4 plays an essential role in the treatment of fatty liver and insulin resistance diseases and has an excellent potential to promote WFB [162,163].

#### 4.4.3. Leptin

Leptin, encoded by the ob gene, is a satiety factor, which is secreted by WAT. Its primary function is to suppress appetite. Studies have shown that the administration of leptin while restricting food could enhance fat catabolism by inhibiting the expression of PPARγ protein [165]. There are many reasons for leptin resistance [166], two of which are the widely recognized ones, including the inability of leptin to reach target site and inhibition or destruction of the intracellular cascade of leptin. Leptin can regulate the expression level of UCP-1. The rats injected with leptin could increase the expression levels of UCP-1 and UCP-2 in the WATs to varying degrees. The study reported that the addition of leptin to the WAT cell culture medium increased the mRNA level of UCP-2 by 63%. Leptin might reduce fat by promoting the expression of UCP proteins [167]. Lin et al. [168] showed that, in addition to regulating the expression of brown fat marker genes, leptin might also mediate the Sh2b1 neuron of the sympathetic nervous system and participate in the activation of brown fat thermogenesis. After inhibiting the leptin receptors, Sh2b1 is responsible for the sympathetic-driven heat production in brown fat, while significantly weakening the other functions.

#### 4.4.4. Catecholamine

Catecholamines are neuronal compounds, containing catechol and amine groups, such as norepinephrine, epinephrine, and dopamine, which are produced from tyrosine as a precursor. Catecholamines bind to G protein-coupled receptors [169]. Catecholamines are induced by physical threats, fight-flight reactions, cold exposure, or other excitements. They bind to ARs in various tissues and activate many metabolic changes, such as lipolysis and heat, increasing the energy consumption. The catecholamines-induced browning is the primary way to reduce the storage of Triacylglycerol (TAG) in the adipose tissues. The specific mechanism of action is as follows. Catecholamines bind to the G protein-coupled β-AR and activate adenylate cyclase (AC), increasing the cAMP. This activates the protein kinase A (PKA), resulting in the phosphorylation of hormone-sensitive lipase (HSL) promotes the breakdown of triacylglycerols. In addition, up-regulation of β3-AR expression promoted white fat browning [170]. In summary, catecholamines hydrolyse triglycerides through β-ARs and PKA signalling.

#### 4.4.5. Fibroblast Growth Factor 21

Fibroblast growth factor 21 (FGF21) is an essential regulator of metabolic processes. Initially, it was reported to stimulate the insulin-independent glucose uptake of adipocytes and inhibit the adipogenesis by inducing the expression of GLUT1. Subsequent studies demonstrated that the cold stimulation and activation of β3-ARs could increase the expression of FGF21 in WAT, suggesting its role in resistance to cold stimulation [171]. The fat derived FGF21 could act in an autocrine/paracrine manner, thereby upregulating the expression level of UCP-1 and other thermogenic genes in adipose tissue. This indicated that FGF21 could promote the recruitment of beige fat cells. Subsequent studies demonstrated that the thermogenic effect of FGF21 on the adipose was mediated by PGC-1α [62]. Interestingly, FGF21 enhanced browning by stabilizing the PGC1α protein without affecting its mRNA expression level, indicating that FGF21 could regulate PGC1α at post-transcription level [172]. However, the activation of FGF21 could also upregulate the expression levels of PPARα/γ, regulating the white adipogenesis. Therefore, the exact role of FGF21 in regulating the lipolysis remains controversial.

#### 4.4.6. Bone Morphogenetic Protein 9 (BMP-9)

Bone Morphogenetic Protein 9 is a novel cytokine, which was cloned from the foetal mouse liver cDNA and belongs to the transforming growth factor-β (TGF-β) superfamily. It is secreted by liver and performs a variety of physiological functions. It can induce in situ and ectopic bone formation, influence the differentiation of cholinergic neurons and proliferation of hematopoietic progenitor cells, and regulate the function of the reticuloendothelial system in the liver, angiogenesis, and tumorigenesis [173]. A recent studies reported that BMP-9 could play a key role in the fat metabolism and other related processes [174]. Specifically, it can inhibit liver gluconeogenesis, convert WATs into BATs, and promote muscle glycogen. Furthermore, it can also synthesize and regulate glucose and lipid homeostasis in the body, increase the uptake and utilization of glucose by muscle tissues, enhance the sensitivities of liver and adipose tissues to insulin, promote insulin synthesis and secretion, inhibit the deposition of lipids in liver, and exert the leptin-like effects. In addition, there are reports that BMP-9 could also enhance the expression of FGF21, thereby inhibiting obesity; FGF21 promotes the recruitment of beige adipocytes by upregulating the protein expression level of PGC1-α [175]. However, given the comprehensive role of BMP-9, its functions and mechanisms should be further explored.

#### 4.4.7. Telmisartan

Telmisartan is a well-known anti-hypertensive drug, which blocks the angiotensin 2 receptor. It has also partial PPARγ agonistic activity and induces insulin sensitivity. Telmisartan could induce the expression of M2 markers in the mouse macrophages in a concentration-dependent manner. The PPARγ plays a key role in the polarization of M2 macrophages and WFB, indicating that telmisartan could promote M2 macrophage polarization by upregulating the expression level of PPARγ. A study reported that M2-polarized macrophages play a key role in the browning of WAT via the activation of type 2 cytokine production during cold exposure [176]. Subsequent investigations reported that telmisartan could increase catecholamine and mRNA levels of intracellular tyrosine hydroxylase (TH), which were the key factors, inducing WFB and promoting the fat metabolism [177].

#### 4.4.8. Irisin

Irisin, reported in an article published in the journal Nature in January 2012 [33], was discovered by researchers as a novel brown fat and fat-transforming hormone. Irisin is a muscle cytokine induced by exercise. It is derived from one of the downstream factors of PGC-1α [33]. After shearing and modification, the fibronectin type III domain (FNDC5) is proteolytically cleaved, producing irisin. Its circulation in the adipose tissues can induce the WFB by activating the ERK and p38MAPK signalling pathways and increasing the expression level of UCP-1. These effects lead to an increase in the mitochondrial respiration and metabolism of adipose tissues [178]. In addition, irisin can also inhibit adipogenesis during differentiation, thereby inhibiting the formation of new adipocytes. At the same time, it can also promote osteogenic differentiation, increase the gene expression levels of HSL and ATGL for lipid metabolism [179]. It can also downregulate the expression of pro-inflammatory cytokines and upregulate that of anti-inflammatory cytokines, thereby reducing insulin resistance and other obesity-related syndromes and inhibiting the phosphorylation and activation of nuclear factor-κB (NF-κB). It could also induce the alteration of the adipose tissue macrophages from M1-like (pro-inflammatory) phenotype to M2-like (anti-inflammatory) phenotype [180]. Studies on the function and mechanism of this regulatory factor have highlighted its importance. Studies have also reported that irisin had different regulatory mechanisms for subcutaneous and visceral fat. Therefore, the mechanism of its function needs to be explored. In mice experiments, IRISIN could increase the expression levels of UCP-1 and CIDEA in the white adipocytes. Similarly, the increased expression levels of IRISIN could also increase energy consumption, reduce body weight, and improve diet-induced insulin resistance [31]. A related study reported that, after 12 weeks of exercise, the levels of PGC-1α and FNDC5 (fibronectin type III domain containing 5) in prediabetes patients and healthy controls slightly increased [64]. The expression levels of UCP-1 and PRDM16 in the WAT and the IRISIN levels in the serum of the two groups showed no significant changes. The study indicated that the long-term exercise had little regulatory effect on the human WFB [65]. This might be due to the species-to-species differences in the regulation of WFB by exercise.

#### 4.4.9. Prostaglandin (PG)

Prostaglandin is a type of eicosanoids derived from PUFA, having prostanoic acid as its basic structure, and is widely distributed in various tissues and fluids in the body. It can bind to receptors and mediate the proliferation, differentiation, and apoptosis of the cell. It plays a key role in regulating the female reproductive function and childbirth, platelet aggregation, and balance of the cardiovascular system [181]. Cyclooxygenase (COX)-2 is a rate-limiting enzyme in the synthesis of prostaglandin and a downstream effector of β-adrenergic signal transduction in WAT [182]. Cold stimulation could induce COX-2/PG pathway, playing a role in the formation of BAT and WFB. Prostaglandin could also differentiate the mesenchymal progenitor cells into brown adipocytes, increase systemic energy expenditure, and protect the mice from HFD-induced obesity [183]. It was specifically involved in the signal transduction of PG/PTGIR/PPARγ pathway and promoted the differentiation of activated mesenchymal progenitor cells through PKA-C/EBPβ pathway to a beige fat cell and increased sensitivity to noradrenaline (NE) [184]. In addition, the COX-2/PG axis also plays a key role in regulating inflammation in the adipose tissues and insulin resistance caused by obesity. Prostaglandin can be prepared by biosynthesis or total synthesis and used as drugs in clinical applications. They play a key role in the treatment of atherosclerosis and thrombotic diseases [185]. Furthermore, they can provide strategies for the improvement of BAT activity and WFB, thereby preventing energy consumption and causing weight gain.

#### 4.4.10. Atrial Natriuretic Peptide (ANP)

Atrial Natriuretic Peptide, also known as an atrial natriuretic peptide, is a cardiac hormone, composed of 28 amino acids, which is synthesized and secreted by cardiomyocytes [186]. It has various cardiovascular and metabolic properties, such as vasodilation, natriuresis, and inhibition of the renin-angiotensin-aldosterone system [187]. In addition, it can also induce lipolysis, lipid oxidation, and fat cell browning, improve insulin sensitivity, and play a key role in the cardiovascular and metabolic homeostasis. Subsequent studies showed that obese people had lower circulating levels of ANP as compared to the healthy people, which was directly related to the inhibition of lipid metabolic processes. The ANP could positively affect lipid metabolism, induce lipid mobilization and oxidation, enhance insulin sensitivity, and increase the energy consumption [188]. The administration of ANP in plasma could increase the plasma adiponectin levels. Adiponectin can suppress appetite and enhance lipids metabolism [187]. The ANP could also inhibit the production of inflammatory cytokines and reduce insulin resistance while inducing the browning of fat cells and mitochondrial biogenesis [189]. The ANP could upregulate the expression of UCP-1 and other browning-related genes, thereby increasing mitochondrial oxidative metabolism and fat oxidation [190].

#### 4.4.11. Mammalian Target of Rapamycin Complex 1 (mTORC1)

The mammalian target of rapamycin (mTOR) is a 250-kDa conservative Ser/Thr kinase, which plays an essential role in regulating the proliferation and differentiation of cells, metabolic recombination, homeostasis, and tumorigenesis. It is an essential signal for the transduction of cellular signals [191]. The well-characterized downstream target of the mTOR complex 1 (mTORC1) is p70 ribosomal S6 kinase 1 (S6K1), which is highly induced by insulin Moreover, the activation of S6K1 by β-AR is dependent on PKA. β-adrenaline can stimulate the activation of the mTORC1/S6K1 pathway via activate PKA and induce the browning of fat cells. In addition to UCP-1 and mitochondrial genes, the key components of the machinery necessary for the metabolism of brown and beige fat cells are also regulated by mTORC1. The activation of mTORC1 by β-adrenergic promotes white adipose browning by upregulating TFEB to promote the expression of PPARA and ERRα (Errα was identified as a transcription factor co-activated by Pgc-1α, required for the high level of oxidative capacity characteristic of brown adipose tissue) as well as several downstream target genes including Pdk4, Cpt1b, Fabp3 and Elovl3 [192].

## 5. White Fat Browning and Resistance to Obesity

Obesity has become a major public health problem worldwide [193], caused by genetic, environmental and social factors, with over two billion people worldwide currently obese. The excessive accumulation of visceral fat caused by obesity can lead to many chronic diseases, such as type 2 diabetes, atherosclerosis and polycystic ovary syndrome [194]. Obesity is caused by a chronic excess of energy intake, which is converted into triglycerides and stored in the adipose tissue [195]. Therefore, reversing obesity is about reversing the energy metabolic balance of the obese body. Most of the previous research in the treatment of obesity has focused on reducing the energy intake of obese individuals. However, as most obese people have insulin resistance which promotes appetite, restricting energy intake to achieve weight loss often leads to weight loss failure. In recent years, many studies have turned to investigating how to enhance the body’s energy expenditure for the purpose of weight loss [196,197]. Brown adipose tissue, an adipose tissue that oxidises fatty acids and thus burns energy, has emerged as a hot topic of research in this area [198]. In addition, beige adipocytes, which are of a completely different origin to brown adipose progenitor cells, have a similar function (Figure 1). Studies have shown that obese patients have significantly lower levels of brown fat and are less active than people with a normal BMI [41]. Therefore, the promotion of browning of white fat and the activation of brown fat activity in the body has become a focus of research in the field of resistance to obesity [197]. Brown adipose tissue is one of the most insulin-sensitive tissues and therefore enhances the body’s uptake of glucose to produce ATP, which can alleviate the ATP deficiency caused by oxidative phosphorylation coupling in the mitochondria [35]. It has been shown that cold stimulation can promote browning of white adipose to counteract the disruption of glucolipid metabolism in obese mice [56]. In addition, the transplantation of brown fat plays an active role in combating obesity and improving insulin sensitivity in mice [199]. In addition, transplantation of beige fat from humans into mice also improved energy metabolism in mice [200]. Interestingly, a recent study has shown that local heat therapy also activates the thermogenesis of beige adipose tissue to improve the body’s energy metabolism and combat obesity [201]. Recent studies have demonstrated that the effect of beige adipose tissue on energy metabolism in mice is less than the contribution of brown fat [202]. Therefore, a better understanding of the activators of brown fat as well as the browning agents of white adipose tissue is important to improve the disorders of glucolipid metabolism in obese patients.

## 6. Summary and Outlook

Fat is an essential tissue in the living body, which can react and change according to the nutrient supply and changes in environmental temperature [28,31,203]. The beige adipocytes are produced by precursor cells in the process of adaptation to cold stimulation. After adaptation, they lose their original functions and acquire CD137, TBX1, and TMEM6 gene expression pattern of WATs. These cells are involved in adaptation to external stimuli, such as cold, or an increase in other physiological heat production activities, excitation of sympathetic nervous system, activation of β3-AR signalling pathway, and complex hormonal responses [42]. In contrast, the activation of mitochondrial-related genes can lead to an increased organelle density of the adipocytes, enhanced mitochondrial function, and improved fatty acid oxidation, thereby restoring their multi-compartmental morphology and beige adipocyte-specific gene expression profile. Meanwhile, the WFB and the de-novo differentiation of beige adipocytes can coexist and are relatively independent. The mutual conversion of beige and white adipocytes to the alternating cold and warm environment involves changes in the expression of different proteins. The changes in the morphology, function, and metal ion concentration of adipose tissues might explain the homology of white and beige adipocytes with different functions.

The beige and brown adipocytes have significant differences in their transcriptomic expression profile. Beige adipocytes have specific Tnfrsf9, Tmem26, HOXC8, HOXC9, CITED1, Eva1, Hspb7, Pdk4, EPSTI1, LHX8, ZIC1 [7,42], which are unique to brown adipocytes. Brown adipose tissues, a major heat-producing adipose tissue, show differences due to species, age, and other factors. Therefore, the characteristics of beige fat adipocytes and their differences from white and brown adipocytes are important for the regulation of lipogenesis and metabolic regulation, which should be explored in the future studies.

White fat browning is a complex process, which is regulated by various nutrients and metabolites and affects the related signalling pathways. The WFB is regulated by UCP-1, PPARγ, PRDM-16, and PGC-1α [108]. The classical theory of WFB is the excitation of sympathetic nervous system and activation of the β3-ARs signalling pathway induced by PRDM-16 and SIRT1, which exert deacetylation effects on PPARγ. The latter process is dependent on the acetylation effect of PRDM-16 and SIRT1 on PPARγ. It can also induce the transcriptional co-activator PGC1α. The translocation of PGC1α into the nucleus and its binding to the promoter regions can activate the genes, such as NRF1/2, TFAM, and SIRT3, thereby regulating the mitochondrial biogenesis and function [84]. In addition to the sympathetic nerves, cellular energy-sensing is also a driving force, which regulates the transcriptional of browning-related genes. For example, AMPK can inhibit fat deposition and induce WFB by downregulating the ACC expression and inhibiting gluconeogenesis [124]. Therefore, it is theoretically possible to upregulate the expression of thermogenesis-related genes by inducing and activating the production of beige fat to treat obesity and other metabolic disorders.

In addition, gender specific factors can affect the WFB, resistance to obesity, and thermogenesis. Although both regulate the expression of UCP-1, PPARγ, PRDM-16, PGC-1α, and other regulatory factors, the effects of nutrients, such as anti-inflammatory, anti-cancer, and anti-oxidation substances, on metabolism are more extensive. It has certain advantages or disadvantages for other metabolic pathways besides lipid metabolism. The signal factors are aimed at targeted therapy and regulate one or more signalling pathways. However, these factors might not be evident for the treatment of obesity and other metabolic syndromes, such as diabetes, heart disease. Both provide extensive study ideas and basis for researchers to treat obesity and other metabolic disorder by inducing the WFB.

Obesity is often accompanied by diseases, such as insulin resistance, type 2 diabetes, and hypertension. Reducing the occurrence of obesity is an effective measure to solve the obesity-related metabolic diseases [2]. Some studies have found that the obese or diabetic patients have low levels of brown fat and beige fat. Therefore, a new method for increasing the ability of adaptive heat production and activation of beige fat cells is needed, which should have beneficial effects on obesity, insulin resistance, and hyperlipidaemia. In recent years, the WFB discovery has provided a new direction for obesity studies. Clarifying the role of relevant transcription factors and signal pathways in WFB and corresponding regulatory mechanisms might help design a treatment strategy for the prevention and treatment of obesity. Providing new theoretical and experimental evidence, as well as new research and development clues for the clinical treatment of obesity, are undoubtedly of greater social significance. Deeper and multi-dimensional studies are needed to investigate the mechanism of WFB.

## Figures and Tables

**Figure 1 ijms-23-07641-f001:**
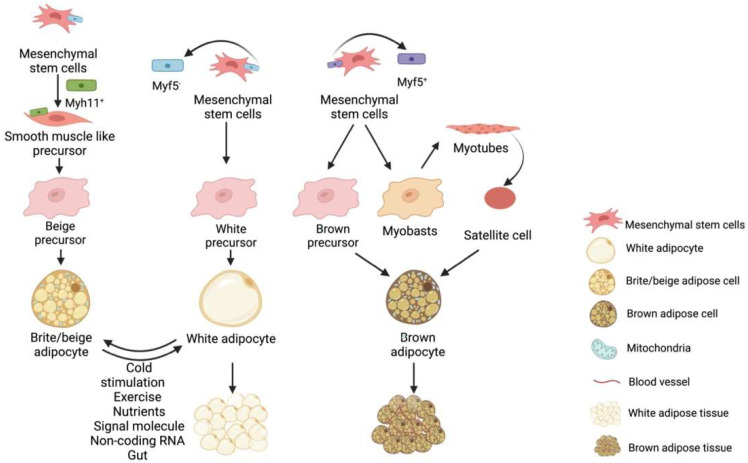
Brown fat, beige fat and white fat precursor cells are derived from different mesenchymal stem cells. Beige adipose precursor cells are derived from Myf5^−^ derived mesenchymal stem cells differentiated into Myh11^+^ positive smooth muscle stem cells. White adipose precursor cells are derived from Myf5^−^ mesenchymal stem cells, while brown adipose precursor cells are derived from Myf5^+^ mesenchymal stem cells. In addition, muscle satellite cells can also differentiate into brown adipocytes under certain conditions. Under certain conditions, white fat and beige fat can be converted into each other.

**Figure 2 ijms-23-07641-f002:**
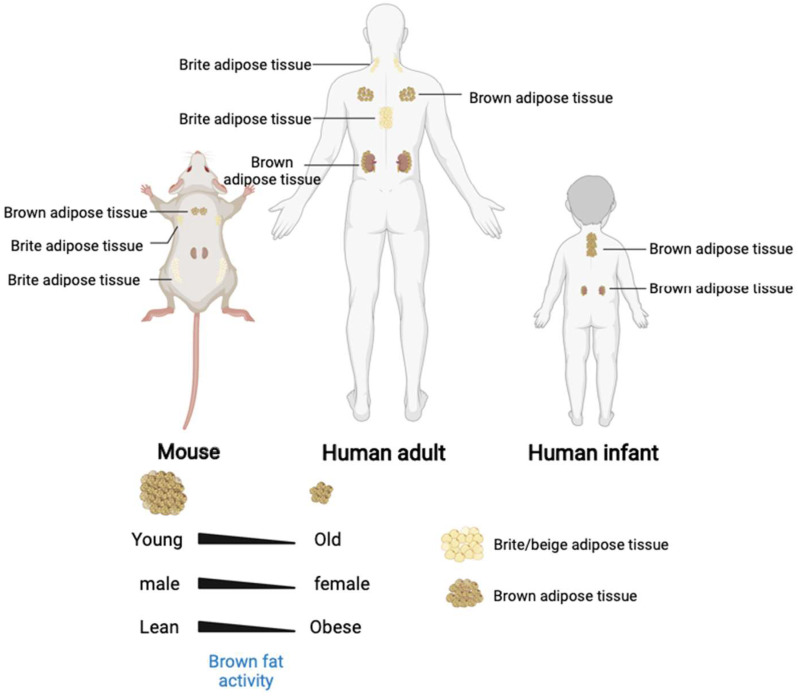
The brown and beige adipose tissue distribution in human and mouse. In addition, the brown adipose tissue activity and quantity decreased with age and associated with gender and stature.

**Figure 3 ijms-23-07641-f003:**
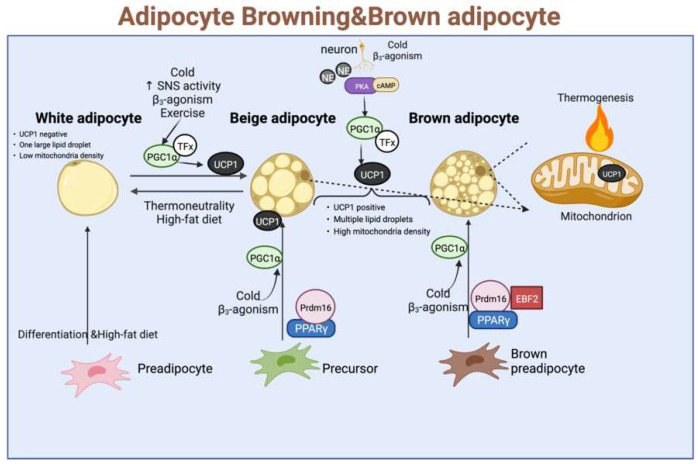
The classic regulatory model for the development of brown fat, beige fat, and white fat, as well as thermogenesis, involves stimulation by external conditions such as cold stimuli, catecholamines, norepinephrine, and exercise. TFx: Trans factors X, where UCP-1 is activated and enters the inner mitochondrial membrane to expend energy in the form of heat dissipation.

**Table 1 ijms-23-07641-t001:** Characteristics of brown, beige, and white fat cells.

	Brown	Beige	White
Morphological	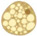	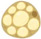	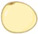
Origin cells	Myf5^+^ cells	Myf5^−^ cells	Myf5^−^ cells
Transcription factors	C/EBPβ; EBF2; PRDM16; UCP-2; PPARγ; PGC1α	C/EBPβ; EBF2; PRDM16; UCP-2; PPARγ; PGC1α	ZFP423; PPARγ
Marker genes	UCP-1; PGC1α; Dio2; Cidea; PPARα; Cox8b; Ppargc1a	UCP-1; PGC1α; Dio2; Cidea; PPARα; Cox8b; Ppargc2a; CD137; TMEM26	Leptin; Fabp4; PPARγ; C/EBPβ
Activators	Cold; β3-AR; Exercise; NP; TH; FGF21; Bmp7; Bmp8bIrisin	Cold; β3-AR; Exercise; NP; TH; FGF21; Irisin

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
