# Peer review of "Factors Associated with White Fat Browning: New Regulators of Lipid Metabolism"

_ijms, 2022, doi:10.3390/ijms23147641_

Round 1

Reviewer 1 Report

The manuscript by Zhang et al. provide an overview of the recent advances in browning white adipose tissue that may show great promise as a therapeutic strategy for the treatment of metabolic complications of obesity. This review article cover interesting topic that has seen significant development or progress in recent years. The main text accurately summarize the current state of understanding of the theme addressed.

Nevertheless, the review may be clearer and more useful to the readers if the Authors will clarify these points:

Figure 1. Does the white adipocyte have no mitochondria? Figure 1 is not fully explained in the text. Please, cite it in more sections in the manuscript (for instance on page 3 line 95).

3.2 Exercise. Could the intensity of exercise, as measured by VO2 max, lactate threshold, or maximum power output, or the duration of physical activity have a different effect on the browning of WAT? Is there any data?

3.3.2 Resveratrol. The authors describe resveratrol as a protective compound with its anti-inflammatory and anti-oxidant potentials, departing from the main topic of the WATB. I recommend the authors should shortened the volume of contents with focusing on the specific theme.

3.4.1 Glucocorticoid (GC). Add “s” glucocorticoids instead of glucocorticoid. The authors state that GSs increase the synthesis of triglycerides, inhibit lipolysis, and promote the accumulation of fats, causing metabolic disorders in the body (page 11, line 527). Please check this information as 95% of glucocorticoids’ hormonal activity is due to cortisol (the rest is due to corticosterone). Cortisol stimulate lipolysis in adipose tissue, thus releasing free fatty acids into the blood (indirect effect, enhance the action of adrenaline and growth hormone).

3.4.8 Irisin. The authors describe this hormone in the 3.2. section. Please provide the available data in one paragraph.

In the abstract and introduction, the authors state that WATB is a potential therapeutic method in the treatment of obesity and its metabolic complications, but I have not found any paragraph on this topic throughout. Please add the appropriate section to your manuscript.

I suggest that authors may summarize the take home message in a diagram highlighting the different factors play and their role in regulating the WAT browning.

Author Response

请参阅附件。

Reviewer 2 Report

IJMS review  June 2022

This is a timely review from Zhang and colleagues on an important and interesting topic. However, I have noted many problems with the manuscript that the authors should address before publication. I have laid out my numerous concerns and questions in the following points.

Line 54 -  “conversion of triglycerides into fat[14]”. Do you mean triglyceride storage in lipid droplets?

Figure 1 – the information here on the developmental origins of white and beige adipocytes does not appear to align completely with the information in reference 22. Can the authors provide more information on the evidence underpinning their diagram? Also, how does this information fit with the recent paper from Westcott et al in Cell Reports DOI: 10.1016/j.celrep.2021.109388 ?  It would also be useful if figure 1 was referenced in all the paragraphs where it is relevant e.g. descriptions on the origins of brown and beige adipose tissue on pages 3 and 4.

Line 77 – “White adipocytes have a single large lipid droplet and fewer blood vessels than the brown adipocytes (Supplement Table 1)”, However,  adipocytes are cells and not tissues and hence do not contain blood vessels. This terminology needs to be carefully corrected in the manuscript.

Line 82 “and consumes heat energy[25]” UCP1 does not consume heat energy. This statement does not make sense.

Figure 2 – the authors should provide supporting evidence and references for the distribution of different fat types in the human infant in the preceding  paragraph a the top of the page as it is not clear where this information derived from. The paragraphs after figure 2 are more relevant to figure 1.  Perhaps the structure and order of the paragraphs and figures could be looked at so that they are in a more logical order?

Lines 115-116 – The meaning of this sentence is not clear.

Line 117 – when what is “induced” ? Again the meaning is not clear.

Line 120 - “alleviate metabolic diseases caused by the excessive WAT deposition[39]” this sentence  is mostly speculation and based on a previous review article. More specific evidence is needed for the claims in this sentence especially re. human disease.

Line 133 – should this be by “by β-oxidation in mitochondria”?

Line 141 – What is FDG?  Also an abbreviation list is required  for this manuscript.

Line 150 – what are these “subregions” and how are they “separated”?

Line 187 – this should be “discovered”.

Line 188 – reference needed for IRISIN information.

Section 3.3 nutrients. Some brief introductory comments are needed here to explain the extent to which why nutrients or pharmaconutrients are relevant to consider at this point in the review.

Line 213 -214 – why is the molecular formula for resveratrol provided and not for other molecules in this section? Please check the information in this opening resveratrol section for accuracy. The description of resveratrol provided here is very unusual.  The statement “The RSV contained in the red wine can protect the body[63]” is too simplistic. I recommend that this paragraph should be rewritten. 

Line 224 – please include references here.

Lines 221 -233. There is a series claims of the health benefits of resveratrol. Some critical insights are needed here. Are these randomised control trials? How credible are these claims? Are these good quality studies? Also, what is the relevance of this section to the title of the review?

Line 235 – “RSV is an agonist of SIRT1” but you start this section by stating that “Resveratrol (RSV; molecular formula, C14H12O3) is a deacetylase activator of anthra- 213 quinone terpene polyphenols”?  Why do you mention pterostilbene here? Again, this whole section needs to be looked at thoroughly and rewritten.  

Line 255 – please can you explain what you mean by “essential” in this context?

Line 265 – please correct “pre-adipocytesinto”.

General point on nutrient section: there needs to be more synthesis and critical insights into possible common mechanisms and/or molecular targets of this compounds. As the moment it is mostly a long list of often unconnected facts with mostly unexplained mechanisms from a range of different sources. Furthermore, all the evidence is presented as equally weighted which means there is no evaluation of the strengths or weaknesses of different studies or experimental approaches.

General point – what are the key signalling or gene expression pathways that determine BAT formation as opposed to WAT? A lot of factors are listed as affecting UCP1 expression but how is UPC1 regulated normally? What is the pathway by which adrenergic stimulation can favour BAT formation? Inclusion of some of this information , perhaps in a diagram in the introduction, may help the reader form more of an overview of the area.

Line 319 – important; capsaicin is not an approved drug for the treatment of obesity. This sentence is misleading.

Line 326 – important; curcumin is nor an approved drug for any medical condition. In fact the abstract of  ref 88 cited by the authors states “No double-blinded, placebo controlled clinical trial of curcumin has been successful.”

Line 435 – how can chrysin activate AMPK?

Line 429 – which PLIN protein are you referring to? What is the normal functions of this PLIN protein isoform?

Line 434 – better to be consistent and use either “Cinnamic aldehyde” or  “Cinnamaldehyde”

Line 445 – where are the randomised control trials that show “it can be used as an effective and safe alternative to the anti-obesity drugs”.

Line 457 – which studies are you referring to here?

Line 462 -  taurine is not an amino acid. This section does not demonstrate much understanding about the biochemistry or biology of taurine.

Line 469 – Need to be clarify that this is a study carried out on an animal model of obesity.

Line 474 – please define iWAT.

Line 501  - please provide the clinical control trials that show that rice bran has an anti-cancer or anti-obesity activities . Where is the evidence that rice bran prevents colorectal cancer?  This section is highly speculative and potentially very misleading.

Line 511 – is there a sentence missing here?

Line 526 – instead of ”factors” why not state that  GC are  “steroid hormones” or a “corticosteroid hormones”.

Line 553  -it is not clear how LXR is inhibited in these circumstances, please clarify.

Line 558b-559 – I don’t understand why 3 effects are listed and then you state there is “no  evident effect”.

Line 560 – “NRG4 559 plays an essential role in the treatment of fatty liver”  where is the evidence for this?

Line 581 – “Catechol and amine groups bind enzymatically at sympathetic nerves” - this statement demonstrates a lack of understanding.  Cathecholamines bind to G protein-coupled receptors. Also how is reference 153 relevant here?

Line 590 – is it really correct that activation of HSL promotes BAT formation? How is this basic  summary of lipolysis relevant to the title of the review?

Line 630 – how can polarised macrophages induce WFB?

Line 635  - citation for Nature paper needed here.

Line 643 . Punctuation error, full stop missing.

Line 655 – PG are eicosanoids derived from PUFA but are not “unsaturated fatty acids”.

Lines 655 -658 – references needed for PG functions.

Lines 661 – 669 – no references provided here.

Line 670 – it not clear how “the treatment of atherosclerosis and thrombotic diseases” is relevant to BAT formation or energy consumption and weight gain?

Lines 675 -682 – lots of references need here and indeed the rest of this section. How is this general information on ANP relevant to BAT formation?

Lines 700 - 703 – meaning of this section is unclear.  How do B2 AR activate mTORC1 in this context?

Line 707 – ERRalpha – what is the function of this protein and how is it relevant to BAT formation? Is the argument here that PPARalpha is commonly induced by mTORC1 and that this leads to BAT formation?

Lines 716 -718 – did not understand this sentence as it written.

Line 737 – “widely used in clinical trials” can you expand on this statement?

Section 4 – important section on UCP proteins, SIRT, PPARgamma etc. Should most of the information in this section be moved to the beginning of the review as it contains key information necessary to understand the overall topic?  A table summarising the main functions of these proteins would be  useful to include too.

Line 825 – metal ions are mentioned here for  and are only alluded to briefly before in section 3.3.11 on menthol. Why are metal ions now prominent in the conclusions section?

Line 833 – “specific” what exactly?

Section 5 – recommend that the word “etc.” is not used.

Lines 868 – 878 – the link with b3 AR and the development of anti-obesity drugs isn’t clear here. Why are these 2 topics bundled together in this paragraph?

Section 5 – there is some repetition with preceding sections. Perhaps this section could be shortened and made more impactful?

Author Response

请参阅附件。

Round 2

Reviewer 1 Report

The manuscript has been revised considering my comments. The revision is highly satisfactory.

Reviewer 2 Report

The authors have responded to all my comments and made many changes that have greatly improved the manuscript.